# Alternative Ecosorbent for the Determination of Trihalomethanes in Aqueous Samples in SPME Mode

**DOI:** 10.3390/molecules27248653

**Published:** 2022-12-07

**Authors:** Gustavo Sánchez-Duque, Juan José Lozada-Castro, Emerson Luis Yoshio Hara, Marco Tadeu Grassi, Milton Rosero-Moreano, Jhon Jairo Ríos-Acevedo

**Affiliations:** 1Grupo de Investigación en Ingeniería Hidráulica y Ambiental (GTAIHA), Universidad Nacional de Colombia, Sede Manizales, Carrera 27 64-60, Manizales 170003, Colombia; 2Grupo de Investigación en Cromatografía y Técnicas Afines GICTA, Departamento de Química, Facultad de Ciencias Exactas y Naturales, Universidad de Caldas, Calle 65#26-10, Manizales 170004, Colombia; 3Grupo de Investigación Estudio de Sistemas Contaminantes, Departamento de Química, Facultad de Ciencias Naturales y Exactas, Universidad de Nariño, Ciudadela Universitaria Torobajo, Pasto 522020, Colombia; 4Grupo de Química Ambiental, Departamento de Química, Universidade Federal do Paraná UFPR, Curitiba 83255-976, Brazil; 5Grupo de Investigación en Electroquímica y Medioambiente GIEMA, Departamento de Química, Universidad Santiago de Cali, Calle 5#62-00, Cali 760032, Colombia

**Keywords:** ionic liquid, mineral clay, ecomaterial, binding layer

## Abstract

A new sorbent material based on modified clay with ionic liquid immobilized into an agarose film was developed as part of this study. It was applied to determine organochlorine pollutants, like disinfection byproducts, through headspace solid-phase microextraction-gas chromatography-electron capture detection (HS-SPME-GC-ECD). The disinfection byproducts determined in this study were used as model molecules because they were volatile compounds, with proven severe effects on human health. Their presence in aquatic environments is in trace concentrations (from pg L^−1^ to mg L^−1^). They are classified as emergent pollutants and their determination is a challenge for analytical chemists. The parameters which affected the extraction efficiency, i.e., number and distance between SPME discs, salt concentration, the temperature of extraction, extraction time, and desorption time, were optimized. A wide linear dynamic range of 10–1000 ng mL^−1^ and coefficients of determination better than 0.997 were achieved. The limits of detection and the limits of quantitation were found in the ranges of (1.7–3.7) ng mL^−1^ and (5.6–9.9) ng mL^−1^, respectively. The precision, expressed as relative standard deviation (RSD), was better than 8%. The developed sorbent exhibits good adsorption affinity. The applicability of the proposed methodology for the analysis of trihalomethanes in environmental and water samples showed recoveries in the range of 86–95%. Finally, the newly created method fully complied with the principles of green chemistry. Due to the fact that the sorbent holder was made of agarose, which is a wholly biodegradable material, sorbent clay is a widespread material in nature. Moreover, the reagents intercalated into the montmorillonite are new green solvents, and during the whole procedure, low amounts of organic solvents were used.

## 1. Introduction

The trihalomethanes (THM) present in drinking water are by-products formed from the disinfection process of surface water. The THMs detected most often in drinking water are chloroform (CHCl_3_), dichlorobromomethane (CHBrCl_2_), dibromochloromethane (CHClBr_2_), and bromoform (CHBr_3_). The formation of these compounds occurs through the chlorination reaction with organic matter, which may be abundant in the surface water before being served to the public [1].

In this sense, factors such as pH and T, chlorine, and humic substance concentration favor the formation of THMs. Thus, during the warmer seasons of the year, there is a substantial increase in the concentration of THMs [2].

Since the presence of THMs in drinking water is related to the increase in cases of bladder cancer, rectal cancer, and spontaneous abortion, the maximum contaminant level [MCL] of THMs were set by the environmental agencies. Nowadays, the maximum concentration of THMs is established at 100 µg L^−1^ both in the USA and Europe [3,4].

In this way, the determination of THMs has become an even bigger challenge given the low concentration of the analytes and the increasingly efficient analytical methods. Currently, the determination of THMs is performed by efficient techniques. However, they are expensive and uncompromising with the principles of green chemistry. These include the techniques recommended by the US EPA 501.2 analytical protocol: liquid-liquid extraction, purge and trap preconcentration and GC-ECD detection [5].

In counterpoint to this issue, Parkinson et al. evaluated the use of SPME to determine THMs (CHCl_3_, CHBr_3_, CHBrCl_2_, CHClBr_2_, 1,2-dibromoethane, and 1,2-dichloroethane) in drinking water samples. The limit of quantification LOQ ranged from 8 to 12 µg L^−1^ and the application of the method to water samples allowed for the determination of THMs between 10 and 60 µg L^−1^. Thus, the authors concluded by stressing the relevance of the method developed for both the ease of use of SPME fibers, which do not require sample preparation, which is both expensive and a source of uncertainty when applied to environmental samples [6].

Gouveia et al. evaluated the risk to which elite athletes are exposed to THMs present in Olympic swimming pools. For this, the authors reported that air and water samples were collected to assess the possible presence of THMs. Samples were collected from an indoor swimming pool facility located in Porto (Portugal) between March and July, that is, in seasons with milder temperatures. In addition, the pool has a water-heating system. For this reason, the authors expected to determine a higher concentration of THMs in both water and air. The samples were analyzed by a GC-MS with an HP-5 column (50 m × 0.2 mm × 0.5 mm). At the end of the sampling campaign, the authors reported a mean value of 205.0 µg m^−3^ of THMs in the air samples around the pool. Of this value, 193 µg m^−3^ consisted of chloroform only. Regarding water samples, the average THM value was 37.6 µg L^−1^ consisting only of chloroform. The authors concluded that athletes are effectively exposed to the long-term effects that THMs can cause. In addition, a significant effect of seasonality on the THM concentration in the air around the pool was observed. That is, the colder days demanded the use of water heating and the THM concentration doubled compared to the days when the heater was not turned on [7].

In recent years it has been encouraged to replace expensive analytical techniques for techniques that use low-cost materials which do not impact the environment and which are as efficient as commercial materials. One such alternative material is the clay mineral [8].

Another solvent that has less impact on the environment is ionic liquid. Ionic liquids are versatile compounds and the use of them to replace conventional organic solvents has attracted attention, as they have proven to be as efficient (in terms of low vapor pressure, high thermal stability, solvation, and remarkable abilities), safer, and healthier [9].

In this sense, the combination of both materials (clay minerals and ionic liquid) has the advantage of low cost, low toxicity, and expressive capacity to sorb numerous contaminants in environmental matrices.

Fiscal-Ladino et al. modified montmorillonite (MMT) by adding an ionic liquid based on imidazole quaternary ammonium salts. The combination of these environmentally friendly materials was shown to have a significant ability to sorb low-polarity compounds such as polychlorinated biphenyls (PCB) in aqueous samples. The authors then applied the sorbent material to a rotating-disc sorptive extraction (RDSE) device to determine PCBs. Thus, the recovery was greater than 80%, the RSD < 25%, and the limit of detection LOD ranged from 3 ng L^−1^ to 43 ng L^−1^. The authors concluded by indicating extraction efficiency using alternative materials that can be compared to commercial materials such as polypropylene, C8, or C18. In addition, under the compromised conditions, the new device was applied to wastewater and the results obtained showed satisfactory applicability [10].

Considering such aspects, the use of montmorillonite and ionic liquids as the extraction phase of sampling devices such as SPME greatly potentiates this technique. Commercial fibers, such as PDMS, in turn, have a considerable acquisition cost. The use of alternative ecomaterials may contribute to lowering the price of the device and may even assist in determining a wide range of analytes [11,12,13,14]. The intercalation of montmorillonite with ionic liquid can modulate the selectivity of such analytes. 

Thus, this work aimed to develop an alternative sorptive phase with environmentally friendly characteristics to be applied in the SMPE device to determine THMs.

## 2. Materials and Methods

### 2.1. Chemicals

All reagents presented purity > 99% (sodium thiosulfate, sodium chloride, methanol, and acetone) and standards were purchased from Sigma Aldrich and Restek. A 200 mg L^−1^ stock solution was prepared from a mixed standard of four trihalomethanes, i.e., chloroform, bromodichloromethane, chlorodibromomethane, and bromoform at 2000 µg mL^−1^ (Restek), and methanol, all reagents were of chromatography-grade. The other working solutions (100 µg mL^−1^ until 1 µg mL^−1^) were prepared by dilution from the stock solution with methanol. The standard, stock, and working solutions were stored at 4 °C [15].

### 2.2. Purification of Montmorillonite

The bentonite raw clay was provided by Bentominercol (Colombia). This material was purified by rinsing with ultra-pure water using the drainage method three times during 24 h. Subsequently, the purified clay was dried and ground to 50 mesh (297 µm) under porcelain mortar.

### 2.3. Intercalation of Ionic Liquids into Montmorillonite: Full Characterization

The intercalation process was conducted under conditions reported by Fiscal-Ladino et al. [10]. In brief, 1.0 g of MMT was dispersed in a 3.0 g of a solution of 1-hexadecyl-3-methylimidazolium bromide (HDMIM-Br) in methanol at 13% *v*/*v*, and the dispersion was kept under vigorous stirring for 1 h at room temperature. The mixture was filtered, and the solid material was washed three times with methanol (20 mL each time) to remove the excess ionic liquid, and then three times with distilled water. The solid was dried in an oven for 24 h at 105 °C.

The characterization of the purified and modified clays [10] was conducted in an XRD XPert PANalytical Empyrean diffractometer with radiation of Cu Kα (λ = 0.1542 nm) operated at 30 kV and 15 mA. The thermogravimetry analysis (TGA) readings were taken in TGA Q500 V6.7 Build 203 equipment in the temperature range of 20–800 °C at a heating rate of 10 C min^−1^. The Fourier-transform infrared (FTIR) analysis was performed in Nicolet iS5 Thermo Scientific equipment between 500 and 4000 cm^−1^ in transmittance mode. The scanning electronic microscopy (SEM) analysis was performed in an FEI Quanta 250 microscope (Thermo Fisher Scientific, Waltham, MA, USA).

### 2.4. Agarose Film as a Holder of the New Green Sorbent

The procedure of the immobilization of montmorillonite intercalated with ionic liquid (MMT-IL) onto agarose gel was performed according to Loh et al. and Hara et al. [16,17].

The sorptive disc was prepared in two steps. In the first place, the MMT was intercalated with ionic liquid 1-hexadecyl-3-methylimidazolium bromide (HDMIM-Br) according to Fiscal-Ladino et al. [10].

Once the modified clay MMT-IL was obtained, it was immobilized onto a hydrogel by mixing 0.45 g of agarose and 1.5 g of modified clay in 40 mL of distilled water at 80 °C, stirring for 1 h and then 1 h more in the freezer at 4 °C to lead to gelification.

The modified montmorillonite with ionic liquid on an agarose film (MMT-IL-AF) that was formed was then punched into circular pieces of a certain size (5.0 mm ID) using a puncher [16].

### 2.5. Headspace SPME Configuration

The full assembly is shown in Figure 1, the new green sorbent phase based on the intercalated montmorillonite with ionic liquids on an agarose film (MMT-IL-AF) was configured as an SPME device and, due to the high volatility of the target analytes (THMs as the target molecules), was set to headspace mode.

For this, 10 mL of the water sample or THM standard solutions were put into a 35 mL SPME vial with a PTFE cap. The new SPME device was put into the headspace approximately 50% above of the top vial capacity. The water sample (10 mL) was pipetted into a 35 mL sample SPME vial, and a magnetic stirrer was placed into the sample.

A Hamilton syringe was used to pierce the PTFE cap and then pieces (1–4) of the MMT-IL-AF discs were hung alternately separated by silicone septa. The assembly was then dipped into methanol for 2 min followed by deionized water for 1 min to condition the films before putting them into the headspace of the sample solutions for extraction. The SPME vial was immediately sealed with a PTFE cap (Figure 1). After stirring at 1500 rpm for 30 min, the films were removed and sonicated with 5 mL of methanol for 5 min. Then 1 µL of methanol was injected into the GC-µECD system.

### 2.6. GC-ECD Instrument

A Hewlett-Packard HP6890GC System Gas Chromatograph (Agilent Technologies Inc. Santa Clara, California, United States) equipped with an µ-ECD detector was used in this experiment. The detector temperature was set at 280 °C. The column was an Rtx-5 30 m × 0.32 mm ID × 0.25 µm. The oven temperature ramp was as follows: the initial temperature of 60 °C was held for 1 min, then it was increased at 20 °C min^−1^ to 150 °C min^−1^. The run time was 5.5 min.

The injector temperature was set to 250 °C and was used in split injection mode (1:10 ratio). The injection volume was 1.0 µL (to avoid the backflush process). The carrier gas was hydrogen (99.9995% purity) autogenerated by a Precision Hydrogen 100–H_2_ Generator (Peak Scientific Instruments Ltd. Inchinnan, UK) at a flow rate of 1.0 mL min^−1^; the auxiliary gas was nitrogen (99.999%, 30 mL min^−1^) supplied by the Cryogas Co. Bogotá, Colombia [15].

### 2.7. Experimental Design for Headspace Extraction

According to the previous description [15,18] when the headspace technique was applied, there were some variables that could have affected the extraction process, the most important being salt concentration, the distance between discs, extraction temperature, the number of discs, and extraction time, among others. The stirring speed of the liquid phase was constant at 1500 rpm. As you can see there are many factors. For this reason, a complete design of experiments (DOE) 2^5^ was carried out, as shown in Table 1.

### 2.8. Design for Desorption Time with Methanol

Considering that the THMs were desorbed with methanol at the end of the extraction process, a completely randomized design CRD was applied to evaluate the time of desorption. The treatments were carried out in the time that elapsed after performing the sonification for 5 min and before injecting. The selected times were 0, 10, 30, and 60 min. Each treatment was applied to two experimental units to test the ANOVA. The time of sonication was 5 min, which was the time in which the detachment of the disc clay started.

### 2.9. Spiked Aqueous and Real Samples

The samples were collected by adjusting the flow to 500 mL min^−1^, and they were preserved with sodium thiosulfate to stop the reaction between natural organic matter [NOM] and residual chlorine. The samples were analyzed immediately after collection.

Ten milliliters of ultrapure water type I were spiked with 500 ng mL^−1^ of the four trihalomethane compounds: chloroform, bromodichloromethane, chlorodibromomethane, and bromoform. This fortified sample was used for the optimization of the HS-SPME microextraction system. The optimized method was then assayed using the laboratory tap water sample. The laboratory tap water sample was collected from the Research Laboratory GICTA’s group at Caldas University.

## 3. Results and Discussion

### 3.1. Characterization of Raw and Modified Bentonite Clay

The characterization of the raw bentonite clay is presented in Figure 2. The peaks of the XRD pattern indicate that the sample predominantly consists of montmorillonite, substantial amounts of quartz, and feldspar impurities, and also contain low amounts of illite, kaolinite, and gipse (Figure 2A).

As shown in Figure 2B, the quartz peak (Q) was reduced, and the montmorillonite peak (M) was displaced to lower angles 2θ at a d space increment thanks to the success of the purification and intercalation processes, respectively.

The thermal properties of raw bentonite clay are shown in Figure 3, where the TGA curve shows a lost weight of 5.2% at 180 °C, and at 580 °C this increased by an additional 3.9%, because of the loss of water by various means (adsorbed water molecules and chemical bonds of OH^−^ groups) in the raw bentonite clay.

The similarity in the appearance of the FTIR spectra of the raw bentonite clay and the Sigma Aldrich montmorillonite standard means that the raw bentonite clay contents are mainly montmorillonite. Moreover, six bands were detected that correspond to 3614 and 3418 cm^−1^ (OH and the hydration stretching), 1641 cm^−1^ (OH and the hydration flexion), 1051 cm^−1^ (Si-OH extension in the plane), 542 cm^−1^ (Si-O flexion), and 460 cm^−1^ (Al-OH flexion). Otherwise, the evidence that the ionic liquid (1-hexadecyl-3-methylimidazolium bromide) was successfully intercalated is shown by the bands at 2780 cm^−1^, which depict the stretching of the C-H bond (Figure 4B).

The SEM micrographs of the raw bentonite clay (Figure 5) reveal the cluster presence of particles with a narrow distribution size with approximate diameters close to 900 nm.

### 3.2. Design of Experiments (DOE) for Headspace Extraction (Screening)

Table 2 presents the results obtained from the DOE for each compound.

The statistical analysis of the results is represented by the Pareto graphs in Figure 6.

The influence of critical parameters in the microextraction procedure, such as temperature, time, and the salting-out effect was evaluated. Considering that this is an equilibrium technique, and especially for this new SPME configuration, additionally the number of discs and the distance between them were studied to establish the optimal conditions for the one-step extraction and to avoid the shadow effect between the discs during the conditions described in Table 2.

Through the Pareto charts for the four THMs, is possible to see that in all of them, the extraction time factor significantly affects headspace extraction in a direct or combined way. The labeled factors represent the following: factor A is salt concentration; B is the distance between the discs; C is the extraction temperature; D is the number of discs; and E is the extraction time.

The interaction AC presented affects, in a relevant way, the answers for the compounds Bromodichloromethane, Dibromochloromethane, and Bromoform, finding that with a salt concentration of 15%, a 2 mm distance between discs, an extraction temperature of 45 °C, four discs, and an extraction time of 30 min presented the best results (see Figure 7, 13th trial).

### 3.3. Design of Experiments DOE for Desorption Time with Methanol

Table 3 includes the results of the areas for the elution times of the THMs with methanol to complete the headspace extraction process.

In one case, it was not possible to integrate the peak due to the low sensitivity of the ECD for this compound. Moreover, according to Figure 8, it is observed that as the time in the elution process increases the size of the signal decreases, which means that although the THMs are soluble in methanol, losses due to evaporation may occur, so the better time for the elution process is 0 min after to the 5 min of sonication.

Although increasing the temperature improved the analyte extraction, it can also cause analyte loss through evaporation due to the volatility of the compounds (e.g., trihalomethanes). At temperatures higher than 50 °C, the extraction yield was very low; at 25 °C, no good behavior was observed. Therefore, 45 °C was selected as the better extraction temperature for the analysis, as shown in Figure 9B,D,E. In our last work with the same molecules, prolonged extraction time led to substantial deterioration of extraction performance due to the loss of solvent from the sample solution [15,18,19,20].

Therefore, 30 min was chosen as the ideal extraction time (Figure 8 and Figure 9A–C). The increasing addition of salt increased the extraction of the target DBP; higher salt concentrations were not necessary based on our experience and that of other authors because excess ionic strength may affect analyte diffusion into the organic phase. For this reason, 15% NaCl was chosen as the salting-out concentration for all subsequent trials using real samples (Figure 8 and Figure 9C,D,F). The use of four discs and a 2 mm distance between them presented the best conditions in this new SPME configuration. This approach was corroborated using the 3D scatter plot with R software (see Figure 9A,E,F).

### 3.4. Analytical Characteristics of the Developed Method

The analytical characteristics are shown in Table 4. The LOD and LOQ were calculated according to the IUPAC rules (LOD = 3 s_B_/m and LOQ = 10 s_B_/m) where s_B_ is the standard deviation of blank and m is the slope of regression linear of the calibration curve at low-range concentrations. In general, the method exhibited good performance parameters, considering that these compounds are volatile. Although the injection was manual, taking into consideration the entire procedure, the relative standard deviations, except that for chloroform, were below 10%. The precision could be further improved by automation or by using the internal standard calibration mode.

As shown in Table 4, the calibration curves for the THM compounds were determined in the same way as for the real and spiked samples. The calculations for the samples were performed directly via the interpolation of the linear equation of the calibration curves.

Based on the overlapping chromatograms (see Figure 10 and Table 4), this HS-SPME-MMT-IL-AF system extracted, purified, and concentrated all four THM compounds from the spiked water sample.

The results of the recovery tests were assessed using the following equation: $\%R=\frac{C_{s}-C_{ns}}{C_{s}}\times 100$, where Cs is the concentration of the spiked sample and Cns is the concentration of the non-spiked or natural sample. As shown in Table 4, all four of the THM compounds spiked into tap water presented satisfactory signals.

By comparing these results with our last work which used the same molecules but another microextraction system, i.e., a solvent bar with a hollow-fiber SBME [15], we found that, both in terms of the results and based on the principles required for each technique, this new configuration is less sensible, and its EF is considerably lower. However, considering that the new development is simple and lab-made, it shows satisfactory performance with respect to THM regulation, the lower RSD, and the more reproducible recovery study, the HS-SPME-MMT-IL-AF method developed herein is potentially attractive and a good alternative.

### 3.5. Analysis of Water Samples

Under the microextraction conditions that presented the best performance, one water sample was tested: laboratory tap water. The chosen conditions were: salt concentration NaCl of 15% (*w*/*v*); a 2 mm distance between discs; an extraction temperature of 45 °C; four discs; an extraction time of 30 min; and a stirring speed of 1500 rpm.

As shown in Table 4, at low concentrations (ng mL^−1^), the presence of at least one THM was detected (i.e., chloroform) in the laboratory tap and drinking water samples. This result indicated that the presence of natural organic matter (measured as total organic carbon TOC) has a direct influence, and this organic matter reacts with the chlorine used in the treatment plant as a disinfectant for protection against microbiological risks. The low amounts of DBP present a risk to the population served, and more barriers for the removal of NOM, the primary cause of disinfection byproducts, are needed. Figure 10 shows the chromatogram of a spiked sample of water taken from the aqueduct of the city of Manizales, Colombia.

In the laboratory tap water analyzed, (*n* = 6) 20 ng mL^−1^ of chloroform was detected (which is still below the limit) and the other THMs showed values lower than their LODs (see Table 4). In this sample, the TOC and residual chlorine were 0.5 and 1.0 µg mL^−1,^ respectively.

### 3.6. By Comparing with Other THMs Analysis Methods

After review of the new microextraction techniques and the possibilities for using greener sorbents, Table 5 presents the different arrangements for the analysis of disinfection by-products using these new procedures for sample preparation. Some other microextraction methods and their performances in the determination of trihalomethanes in drinking water are shown in Table 5 for comparison with the present work.

From this table, it can be observed that, with some exceptions, the developed method has a higher limit of detection but still acceptable for the target analytes and their regulation, a recovery percentage closer than the target, and lower variability than the other microextraction methods.

## 4. Conclusions

The newly developed method based on HS-SPME configuration using a greener lab-made sorbent phase reached a satisfactory result in comparison to other commercial methods, with better recoveries (>87%), less RSD (<8%), an affordable cost for water sample analysis, and enough LOD and LOQ requirements.

The factors of extraction time, salt concentration, and extraction temperature were identified as relevant factors in headspace extraction using a solid-phase microextraction HS-SPME configuration process for THM isolation from water samples, in strong accordance with other equilibrium extraction techniques, obtaining the best results in 30 min, with a salt concentration of 15% and a temperature of 45 °C.

The ionic liquid intercalated with montmorillonite on an agarose film as a new lab-made immobilized film for microextraction issues has been compiled by four times in the way of the green chemistry ideology. The adsorbent-holder agarose gel is biodegradable, the adsorbent montmorillonite clay is a widespread eco-material, the ILs are the new green sorbents, and the extraction procedure uses a less organic solvent.

Finally, this new lab-made sorbent phase used in an SPME configuration could provide many opportunities in the microextraction field for the analysis of volatile compounds in water.

## Figures and Tables

**Figure 1 molecules-27-08653-f001:**
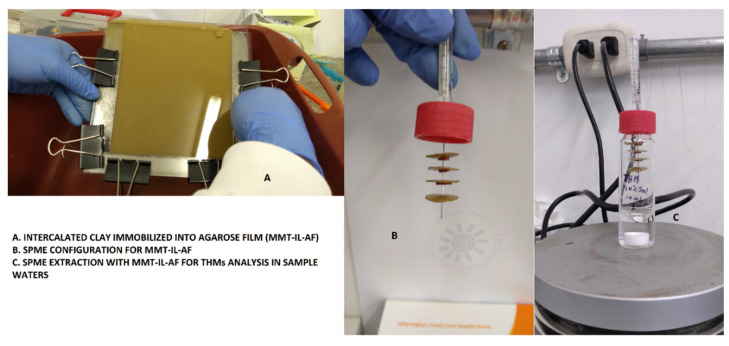
Assembly of montmorillonite intercalated with ionic liquids on an agarose film for THM extraction from water samples as HS-SPME configuration [18].

**Figure 2 molecules-27-08653-f002:**
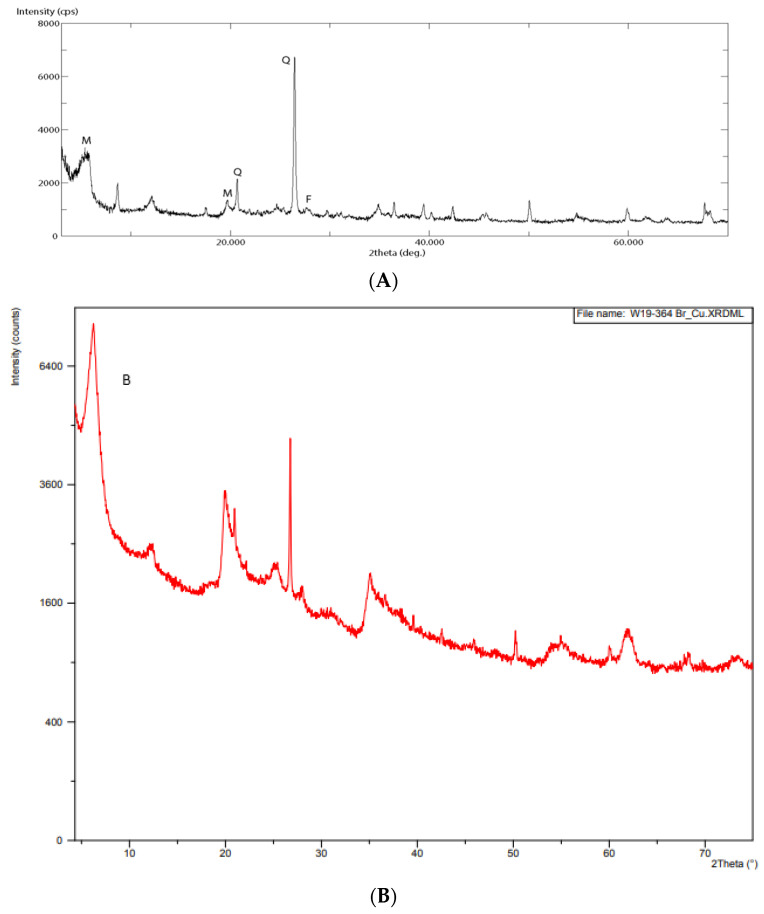
Comparison of X-ray diffraction (XRD) patterns (**A**). The XRD patterns of raw montmorillonite and (**B**). The XRD patterns of modified montmorillonite with ionic liquids.

**Figure 3 molecules-27-08653-f003:**
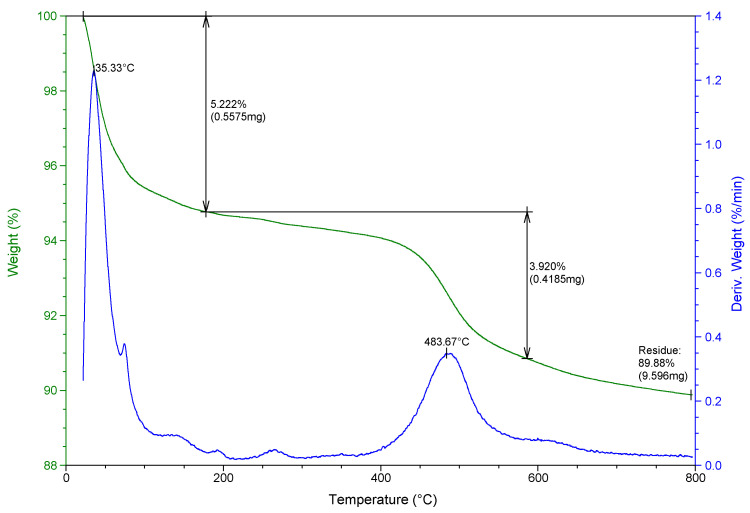
TGA analysis of purified raw bentonite clay.

**Figure 4 molecules-27-08653-f004:**
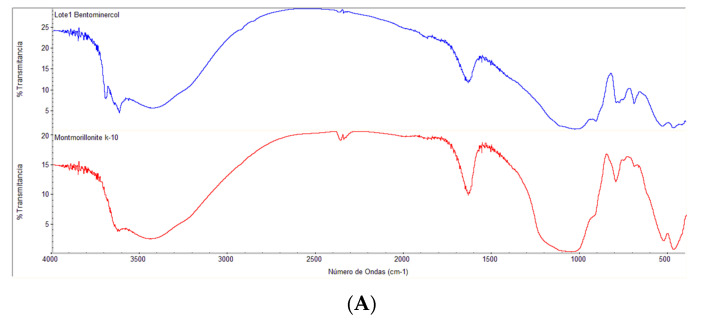
FTIR spectra (**A**) of the purified raw bentonite clay and the Sigma Aldrich K10 montmorillonite standard, and (**B)** the raw (red) and modified (blue) bentonite clay.

**Figure 5 molecules-27-08653-f005:**
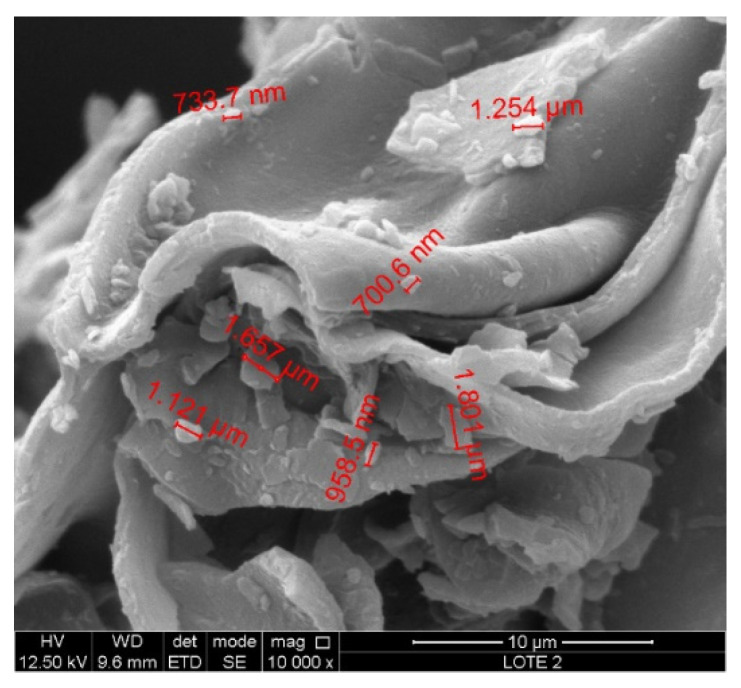
Micrographs SEM of the purified raw bentonite clay.

**Figure 6 molecules-27-08653-f006:**
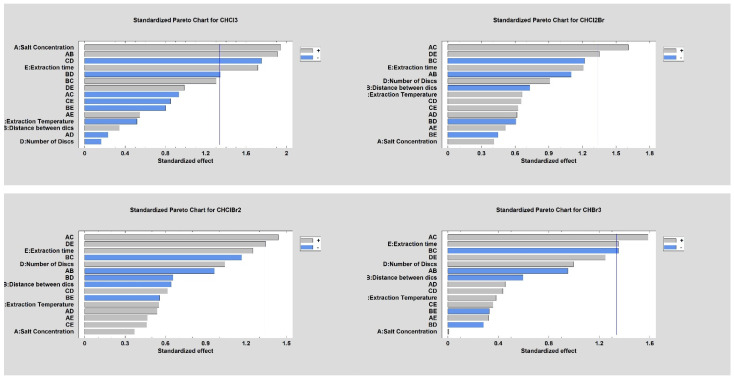
Pareto graphs for the optimization of THM analysis from water sample extraction by HS-SPME.

**Figure 7 molecules-27-08653-f007:**
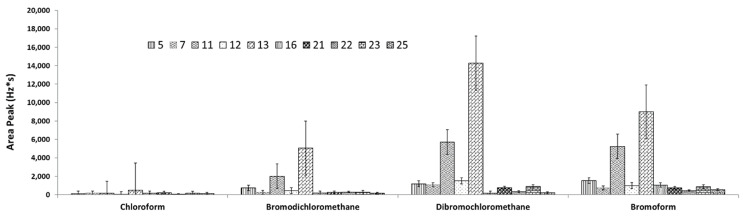
DOE for optimization extraction by HS-SPME showing some representative assays.

**Figure 8 molecules-27-08653-f008:**
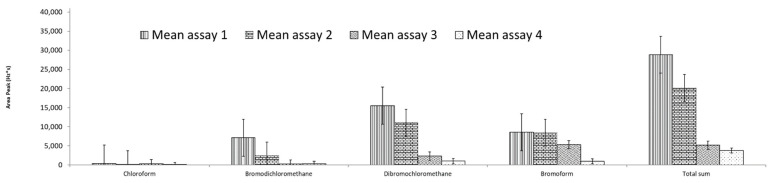
CRD design for desorption time optimization after HS-SPME under sonication with 5 mL of methanol showing the four times assayed and its replicates under the best extraction conditions (13th trial).

**Figure 9 molecules-27-08653-f009:**
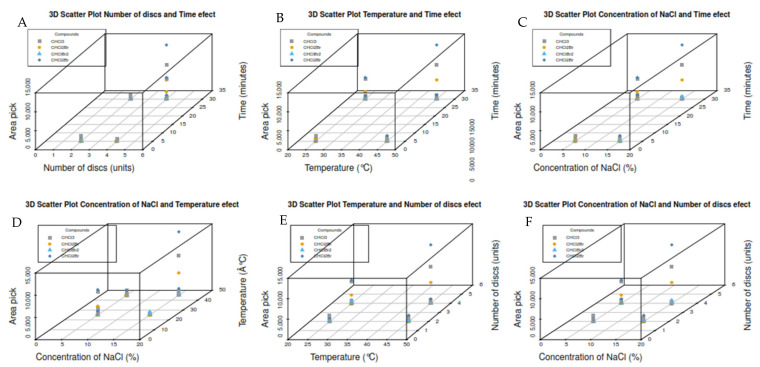
(**A**–**F**) A 3D Scatter plot for optimization of microextraction conditions in the new SPME configuration at 2 mm of the distance between discs.

**Figure 10 molecules-27-08653-f010:**
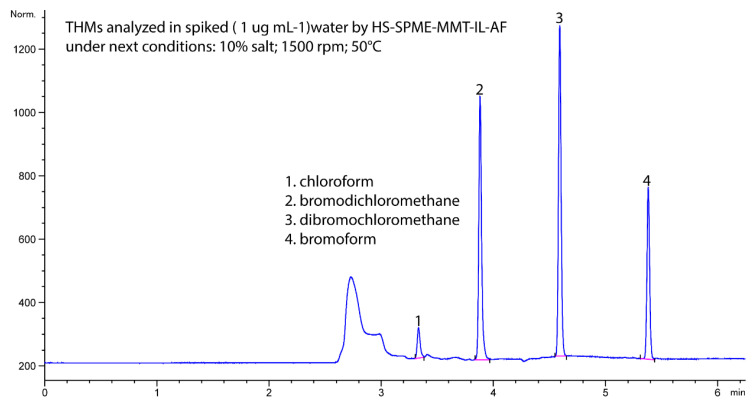
A typical chromatogram of HS-SPME configuration for THM extraction from spiked tap water with HS-SPME-MMT-IL on an agarose film.

**Table 1 molecules-27-08653-t001:** Application of DOE 2^5^ to optimize HS-SPME extraction.

Factor	Low Level	High Level
1 Salt concentration (% *w*/*v*)	5	10
2 Distance between discs (mm)	2	5
3 Extraction temperature (°C)	25	45
4 Number of discs	2	4
5 Extraction time (min)	5	30

**Table 2 molecules-27-08653-t002:** Results of the optimization extraction by HS-SPME by using a new hybrid ecomaterial of MMT-IL-AF as sorbent phase from spiked sample at 500 ng mL^−1^.

Exp.	Factors (Units)					Area Peak			
	1 (%)	2 (mm)	3 (°C)	4 (# discs)	5 (min)	CHCl_3_	CHCl_2_Br	CHClBr_2_	CHBr_3_
1	5	5	45	2	30	167	148	84	204
2	15	5	45	4	5	16	24	36	204
3	5	5	45	4	5	38	25	57	60
4	15	5	25	2	30	412	39	254	328
5	5	2	25	2	5	96	740	1184	1528
6	5	2	25	4	5	193	79	371	342
7	5	2	45	4	30	113	255	1075	732
8	5	2	45	2	5	54	115	477	340
9	15	5	45	4	30	358	18	61	102
10	15	2	25	2	30	147	18	51	176
11	5	5	25	4	30	22	2006	5709	5252
12	15	5	45	2	5	497	440	1507	992
13	15	2	45	4	30	3	5062	14,281	9000
14	15	2	25	2	5	13	14	14	37
15	15	5	25	4	30	171	5	54	254
16	5	2	45	2	30	260	191	172	1058
17	15	2	45	4	5	1	269	156	376
18	15	5	25	2	5	182	91		10
19	5	5	25	4	5	17	52	157	625
20	15	5	25	4	5	235	0	0	0
21	5	5	25	2	30	54	250	787	745
22	15	5	45	2	30	166	293	348	441
23	5	2	25	2	30	61	271	892	878
24	15	2	45	2	5	144	229	119	380
25	5	2	45	4	5	70	151	228	549
26	15	2	45	2	30	98	207	70	229
27	5	5	25	2	5	0	181	109	229
28	5	5	45	2	5	82	0	0	0
29	5	5	45	4	30	89	142	86	262
30	15	2	25	4	30	684	141	358	395
31	15	2	25	4	5	30	0	0	0
32	5	2	25	4	30	260	21	29	52

**Table 3 molecules-27-08653-t003:** Results (peak area) by replicate measurements of desorption time optimization.

THMs	Assay			
	1	2	3	4
	0 min.	10 min.	30 min.	60 min.
Chloroform	497	355	637	20
	327	0	14	41
Bromodichloromethane	5062	2556	174	239
	9179	2338	297	318
Dibromochloromethane	14,281	9572	1088	1723
	16,713	12,582	3548	295
Bromoform	9000	7657	3230	1804
	8119	9212	7350	120

**Table 4 molecules-27-08653-t004:** Analytical characteristics of the developed HS-SPME-MMT-IL-AF system.

Compound	LOD	LOQ	Linear Range	R^2^	RSD	Relative Recovery	Enrichment
	[ng mL^−1^]	[ng mL^−1^]	[ng mL^−1^]		[%]	[%]	Factor *
Chloroform	1.7	5.6	10–1000	0.997	8.3	87 ± 9	87
Bromodichloromethane	1.9	6.2	10–1000	0.997	7.7	89 ± 7	89
Chlorodibromomethane	2.7	8.9	10–1000	0.997	5.7	86 ± 3	86
Bromoform	3.7	9.9	10–1000	0.999	5.8	95 ± 2	95

* Theoretical EF equal to 100 times.

**Table 5 molecules-27-08653-t005:** Developed methods for analyzing THMs in drinking water by using microextraction techniques as sample preparation and greener sorbents.

Extraction	Instrument	LOD	LDR	Relative Recovery	RSD	Ref.
Technique		ng mL^−1^	ng mL^−1^	%	%	
HF-SBME	GC-µECD	0.017–0.037	10–900	74–91	5.7–10.3	[15]
HF-LPME	GC-ECD	0.018–0.049	0.88–337.5	80.3–104.2	1.8–3.7	[19]
HS	GC-MS	0.023–0.102	1.04–230.8	86.3–90.0	6.8–7.8	
HS-SPME	GC-µECD	0.057–0.319	5–200.	74.7–120.9	1.8–11.0	[20]
SBSE	GC-HRMS	N.R.		N.R.		[21]
HS	GC-µECD	0.09–0.14	0.1–100		2.4–4.3	[22]
ITEX	GC-MS	01–10		90–103	<10	[23]
		(pg/mL)				
HS-SPME-MMT-IL-AF	GC-µECD	1.7–3.7	10–1000	87–95	<8	This work

LOD: Limit of detection; LDR: Linear dynamic range; RSD: Relative standard deviation; N.R.: not reported.

## Data Availability

Not applicable.

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
