# Peer review of "Alternative Ecosorbent for the Determination of Trihalomethanes in Aqueous Samples in SPME Mode"

_molecules, 2022, doi:10.3390/molecules27248653_

Round 1

Reviewer 1 Report

The article from Rosero-Moreano et al. reports a new sorptive phase for extracting trihalomethanes in water. The phase consists of natural clay intercalated with an ionic liquid. This composite material is mechanically stabilized using agarose as a binder. The article is interesting, but some aspects are suggested to improve the quality of the manuscript:

- Revise the superscript throughout the entire manuscript.

- Line 29. Substitute “less” with “better”. Precision and RSD are inversely related.

- Some references, like [7], are fully described in the text, while the literature to support the importance of natural materials or green analytical chemistry is scarcely considered.

- Line 123. “n” times? Please specify.

- Lines 175-178. Specify the incubation temperature.

- Why did the authors use only one microliter for GC analysis? The chromatogram (Figure 10) is good enough to increase this volume. The sensitivity could be even better with this strategy.

- Specify the composition of the GC column.

- Is the temperature maintained at 150 ºC for 1 min in the GC? Otherwise, the run time would be 5.5 and not 6.5 min.

- Improve the quality of Figures 2 and 6.

- Although the authors measure the IR spectra under the transmittance mode, I suggest (just a suggestion) representing the data as absorbance values. As an analytical chemist, I prefer absorbance (easily related to the concentration) to transmittance, even in the characterization assays.

- In Figure 10, specify the type of water (tap, natural…).

- Table 4. Substitute “recovery” with “relative recovery”.

- Table 5. Substitute “recuperation” with “relative recovery”.

Author Response

Response to reviewer 1 Comments

Point 1: Check the superscript throughout the entire manuscript.

Answer 1: Superscripts are reviewed and corrected throughout the manuscript.

Point 2: Line 29. Replace “less” with “better”. Accuracy and RSD are inversely related.

Answer 2: The suggestion is accepted and in line 29 “less” is replaced by “better”.

Point 3: Some references, such as [7], are fully described in the text, while the literature supporting the importance of natural materials or analytical green chemistry is barely considered.

Answer 3: the literature that supports the importance of natural materials is complemented.

Point 4: Line 123. “n” times? Specify, lines 175-178. Specify the incubation temperature.

Answer 4: -On line 123. The number of times is specified; - Lines 175-178. The incubation temperature is specified.

Item 5: Specify the composition of the GC column; Is the temperature held at 150°C for 1 min on the GC? Otherwise the execution time would be 5.5 and not 6.5 min.

Answer 5: The composition of the GC column is specified and the temperature ramp of the method is reviewed.

Point 6: Although the authors measure IR spectra in the transmittance mode, I suggest (just a suggestion) to represent the data as absorbance values. As an analytical chemist, I prefer absorbance (easily related to concentration) to transmittance, even in characterization assays.

Answer 6: The authors agree with their assessment and consider it very correct, however, it is decided to leave the % of processing.

Point 7: Improve the quality of Figures 2 and 6. In Figure 10 specify the type of water (tap, natural…). Table 4. Replace “recovery” with “relative recovery”. Table 5. Replace “recovery” with “relative recovery”.

Answer 7: The quality of Figures 2 and 6 is improved. Figure 10 specifies the type of tap water. In Table 4. "Recovery" is replaced by "relative recovery". In Table 5. "Recovery" is replaced by "relative recovery".

Reviewer 2 Report

Interesting studies on the important analytical topic of headspace solid phase microextraction. The material and conditions for the solid phase extraction were developed for analytes from the trihalomethanes group as model molecules for disinfection byproducts. 

The material was developed on the basis of bentonite as a natural sorbent which was modified with an ionic liquid and agarose was used as a natural holder.

The data in the manuscript are valuable however, corrections, additions and clarifications are necessary before publication of the manuscript

1.     Authors should explain in the manuscript why among the numerous ionic liquids, 1-hexadecyl-3-methylimidazolium bromide (HDMIM-Br) was chosen for the intercalation of montmorillonite. Agarose and montmorillonite discs without IL should be used to compare the extraction efficiency. Maybe without an ionic liquid the effect would be the same.? This information is not found in the manuscript.

2.     The authors used purified raw bentonite for the production of extraction discs, but Figures 3, 4, 5 show data for raw bentonite. The XRD and IR of the purified bentonite should be analyzed and compared to show how this material changed or did not change after intercalation. The XRD pattern and IR spectrum of the pure ionic liquid should be also added and compared with the montmorillonite intercalated with the ionic liquid.

3.     The information in the Materials and Methods section is not clear and the experimental conditions are not always the same as those presented in the Results and Discussion section.

a.     the headspace SPME technique involves placing the HS-SPME device above the surface of the analyte solution. Please explain why the section 2.5 (page 4, line 175-178) says that „the assembly… is dipping into the sample solution for extraction”

b.     the authors are inconsequent with the treatment times used in desorption studies. In the section 2.8 (page 6, line 220) authors wrote „The selected times were: 0, 15, 30, and 45 minutes…” while in table  3 the results for 0, 10, 30 and 60 minutes were given.

c.     please add in the Methods section and in the caption of the table 2 what were the concentrations of the analytes in the solution used in the optimization extraction by HS-SPME?

d.     how much methanol was used to desorb the analytes from the discs? In section 2.5, line 180 the authors have written that „the films were removed and sonicated with 100 μL of methanol for 5 min”, while in the caption of figure 8 (line 388-389) there is information about "sonication with 5 mL of methanol" What sonication conditions were used?

e.     the legend in fig. 8 is hardly legible –replacing of symbols in the legend with a specific extraction parameter for which data are presented is necessary

4.     The authors refer to previous research and works without providing a reference – line 206; 397-398, 441

Author Response

Response to Reviewer 2 Comments

Point 1: Authors should explain in the manuscript why among the numerous ionic liquids, 1-hexadecyl-3-methylimidazolium bromide (HDMIM-Br) was chosen for the intercalation of montmorillonite. Agarose and montmorillonite discs without IL should be used to compare the extraction efficiency. Maybe without an ionic liquid the effect would be the same.? This information is not found in the manuscript.

Response 1: As was referenced during the manuscript, in the last our works we have been used this type of IL firstable for it has a simple way to syntetize from methyl imidazole plus 1-bromohexadecane in ethyl acetate medium under inert nitrogen atmosphere during 8 h of reflux and in second order cause it long carbon chain allow to increase the interlaminar space and turn up more hydrophobic the surface [10] {section 2.3.2. Synthesis of the ionic liquid. The ionic liquid HDMIM-Br was synthesized according the procedure described by Min et al. Briefly, the procedure involved mixing equimolar amounts of methyl imidazole (5.0 g, 61 mmol) and 1-bromohexadecane (18.3 g, 61 mmol) in a reaction flask in ethyl acetate (50 mL). The mixture was vigorously stirred at reflux for 48 h, and after this time, the solution was evaporated to remove the solvent and obtain HDMIM-Br as a pale yellow solid. This compound was characterized by 1H NMR and 13C NMR.}

Really in the other our studies by working with ILs the sorption capacities of alone MMT was at least 10% {see referenced our papers with clay: 1-PCBs in Anal. Chim. Acta 2016; 2-Parabens in Journal Sep. Sci. 2017; 3-Ocratoxin A in Sci. Chrom. 2018; 4-Blue methylene and color in Producción+Limpia 2019; 5-PAHs in Marine Polut. Bull. 2020; 6-Triazines in Helyon 2021}, for this strong evidence no were showed and non experimented, in the same conditions as reported other authors. In other our work no yet published results for hormones extraction from water samples, the alone MMT has reached until 33% sorption capacities over lightly polar compounds <see attached the results>

In the other hand nowadays we are measuring (herein isn´t showed) the contac angle proof in a planar model for the modified clay to assure the hydrophobic feature in attention of presence the long carbon chain from IL intercalated into the clay. So the presence of ILs has enhanced the adsorption features and has done wider the range polarity of clays.

Point 2: The authors used purified raw bentonite for the production of extraction discs, but Figures 3, 4, 5 show data for raw bentonite. The XRD and IR of the purified bentonite should be analyzed and compared to show how this material changed or did not change after intercalation. The XRD pattern and IR spectrum of the pure ionic liquid should be also added and compared with the montmorillonite intercalated with the ionic liquid.

Response 2: The reviewer is true, the MMT used was the purified, but along these years have been increased the hypothesis that is not necessary to purify case is a strictly aspect related with spectral purity that not have influence in it performance. In the figures 2B-XRD and 4B-FTIR are shown the material after intercalation and discussed herein an in other our papers. About ILs characterizarion, this aspect really was reached in the paper [10] with other published colleague that by independent way has complete assessment of ILs and that it serves for our conjunction studies <see Montano et al. Materials Chemistry and Physics 2017>

Point 3: The information in the Materials and Methods section is not clear and the experimental conditions are not always the same as those presented in the Results and Discussion section.

  1. the headspace SPME technique involves placing the HS-SPME device above the surface of the analyte solution. Please explain why the section 2.5 (page 4, line 175-178) says that „the assembly… is dipping into the sample solution for extraction”

Response 3a: The reviewer is true, the line will be rewritted as follow: …The assembly was then dipped into methanol for 2 min followed by deionized water for 1 min to condition the films before putting it into the headspace of sample solution for extraction

As show from figure 1 really the extraction is on the headspace mode.

  1. the authors are inconsequent with the treatment times used in desorption studies. In the section 2.8 (page 6, line 220) authors wrote „The selected times were: 0, 15, 30, and 45 minutes…” while in table 3 the results for 0, 10, 30 and 60 minutes were given

Response 3b: The reviewer is true, the tested times for desorption step were those on the table, the line will be rewritted as follow: … The selected times were: 0, 10, 30, and 60 minutes, each treatment was applied to two experimental units to test the ANOVA.

  1. please add in the Methods section and in the caption of the table 2 what were the concentrations of the analytes in the solution used in the optimization extraction by HS-SPME?

Response 3c: The spiked sample concentration has been put on section 2.9 but in the table 2 the caption legend of table was rewritten as follows… Table 2. Results of the optimization extraction by HS-SPME by using a new assemble of MMT-IL-AF sorbent phase from spiked samples at 500 ng mL-1

  1. how much methanol was used to desorb the analytes from the discs? In section 2.5, line 180 the authors have written that „the films were removed and sonicated with 100 μL of methanol for 5 min”, while in the caption of figure 8 (line 388-389) there is information about "sonication with 5 mL of methanol" What sonication conditions were used?

Response 3d: The reviewer is true, the phrase in section 2.5 will be rewritten as follows … the films were removed and sonicated with 5 mL of methanol for 5 min

  1. the legend in fig. 8 is hardly legible –replacing of symbols in the legend with a specific extraction parameter for which data are presented is necessary

Response 3e: We are agree with the reviewer, the legend in fig. 8 has been replaced with numbers in the legend related to a specific extraction parameter (assay) which were merged in one mean value of the two replicates

Point 4: . The authors refer to previous research and works without providing a reference – line 206; 397-398, 441

Response 4: The our proper four references[15,18,19 and 23] have been placed in the requested places

Reviewer 3 Report

Title: Alternative ecosorbent for the determination of trihalome- 2 thanes in aqueous samples on SPME mode

 Article Type: Original paper

 Please uniform units (line 26) mL-1 into mL-1

Introduce abbreviations when first time is mentioned in Manuscript (e.g. LOD/LOQ)

Line 116: is this (100, 1 μg mL-1) typos?

In line 137 the authors claimed that the characterization of purified and modified clays were performed by XRD, TGA, FTIR and SEM, but some of this analysis show only analysis of raw bentonite clay (TGA and SEM). Please change the methodology or add a missing analysis.

Line 149: please add according to whom.

Line 248, Figure 2: diffractograms are not clear and resolution is poor.

Line 397: Figure 9.1, 9.4 and 9.5 does not match to Figure 9. Correct it.

In abstract authors claimed that “the limits of detection and the limits of quantitation were found in the ranges of 1.7 and 5.6 ng mL−1, respectively. Is this for chloroform (Table 4) or for investigated THMs (Table 5) wher LOD is in the range from 1.7 to 3.7.

Author Response

Response to Reviewer 3 Comments

Point 1: Please uniform units (line 26) mL-1 into mL-1

Response 1: It has been done

Point 2: Introduce abbreviations when first time is mentioned in Manuscript (e.g. LOD/LOQ)

Response 2: It has been done

Point 3: Line 116: is this (100, 1 μg mL-1) typos?

Response 3: Really not typos, it were prepared a several dilutions in cascade

Point 4: In line 137 the authors claimed that the characterization of purified and modified clays were performed by XRD, TGA, FTIR and SEM, but some of this analysis show only analysis of raw bentonite clay (TGA and SEM). Please change the methodology or add a missing analysis.

Response 4: All data about the modified and purified raw material is registered in our paper reference [10]

Point 5: Line 149: please add according to whom.

Response 5: It ha been done

Point 6: Line 248, Figure 2: diffractograms are not clear and resolution is poor.

Response 6: The figure 2 ha been enhanced

Point 7: Line 397: Figure 9.1, 9.4 and 9.5 does not match to Figure 9. Correct it.

Response 7: It ha been done

Point 8: . In abstract authors claimed that “the limits of detection and the limits of quantitation were found in the ranges of 1.7 and 5.6 ng mL−1, respectively. Is this for chloroform (Table 4) or for investigated THMs (Table 5) wher LOD is in the range from 1.7 to 3.7.

Response 8: The reviewer I true, it has been corrected

Round 2

Reviewer 2 Report

I recommend the manuscript for publication in its present form.